# Genome-Wide Identification of TCP Gene Family in *Dendrobium* and Their Expression Patterns in *Dendrobium chrysotoxum*

**DOI:** 10.3390/ijms241814320

**Published:** 2023-09-20

**Authors:** Ye Huang, Xuewei Zhao, Qinyao Zheng, Xin He, Meng-Meng Zhang, Shijie Ke, Yuanyuan Li, Cuili Zhang, Sagheer Ahmad, Siren Lan, Zhong-Jian Liu

**Affiliations:** 1Key Laboratory of National Forestry and Grassland Administration for Orchid Conservation and Utilization, College of Landscape Architecture and Art, Fujian Agriculture and Forestry University, Fuzhou 350002, China; 5221726097@fafu.edu.cn (Y.H.); zxw6681@163.com (X.Z.); qinyaozheng@fafu.edu.cn (Q.Z.); 5220422102@fafu.edu.cn (X.H.); 1220428020@fafu.edu.cn (M.-M.Z.); 3210422027@fafu.edu.cn (S.K.); lyy9902140575@163.com (Y.L.); cuilizhang@fafu.edu.cn (C.Z.); sagheerhortii@gmail.com (S.A.); 2College of Forestry, Fujian Agriculture and Forestry University, Fuzhou 350002, China

**Keywords:** TCP gene family, genome-wide analysis, *Dendrobium*, expression pattern

## Abstract

The TCP gene family are plant-specific transcription factors that play important roles in plant growth and development. *Dendrobium chrysotoxum*, *D. nobile*, and *D. huoshanense* are orchids with a high ornamental value, but few studies have investigated the specific functions of *TCP*s in *Dendrobium* flower development. In this study, we used these three *Dendrobium* species to analyze *TCP*s, examining their physicochemical properties, phylogenetic relationships, gene structures, and expression profiles. A total of 50 *TCP*s were identified across three *Dendrobium* species; they were divided into two clades—Class-I (PCF subfamily) and Class-II (CIN and CYC/TB1 subfamilies)—based on their phylogenetic relationships. Our sequence logo analysis showed that almost all *Dendrobium TCP*s contain a conserved TCP domain, as well as the existence of fewer exons, and the *cis*-regulatory elements of the *TCP*s were mostly related to light response. In addition, our transcriptomic data and qRT-PCR results showed that *DchTCP2* and *DchTCP13* had a significant impact on lateral organs. Moreover, changes in the expression level of *DchTCP4* suggested its important role in the phenotypic variation of floral organs. Therefore, this study provides a significant reference for the further exploration of TCP gene functions in the regulation of different floral organs in *Dendrobium* orchids.

## 1. Introduction

In recent years, many studies have shown that *TCP*s are widely involved in plant growth and development, including seed germination [1,2], flower development [3,4,5], leaf development [3,6], gametophyte development [7,8,9], and lateral branching [10]. Previous analyses have shown that TCP is a member of the RHH superfamily, but the crystal structures of the Class-I TCP domain from *AtTCP15* and the Class-II TCP domain from *AtTCP10* in complex with different double-stranded DNA (dsDNA) reveal that the TCP domain is a distinct DNA-binding motif, and the homodimeric TCP domains adopt a unique three-site recognition mode [11,12]. Based on the differences in the TCP domains, the TCP gene family can be divided into two subfamilies: Class-I (PCF or TCP-P) and Class-II (TCP-C) [13]. Class-I has a conserved four-amino-acid deletion in the TCP domain. Class-II is further subdivided into two clades: the CINCINATA(CIN) clade and the CYC/TB1 clade [14].

The distinct roles that *TCP*s play in plant growth and development are the subject of ongoing investigations. For instance, *Fragaria* × *ananassa TCP11* interacts with MYB or bHLH gene family transcription factors to positively regulate the synthesis genes of proanthocyanidins and flavonoids [15]. TCP TFs also play a role in plant abiotic stress alleviation [16]. The CUC genes, which controls the morphogenesis of the shoot lateral organs and regulates the formation of the shoot meristem, is of critical importance to plant development [17]. Moreover, TCP TFs have been extensively studied for their roles in the process of flowering and fruit development. For example, CYC/TB1 genes plays an important role in forming the symmetry of flowers, and CIN genes affects petal development. The deletion of *TCP5* in mutant *A. thaliana* causes wider petals than that of wild-type *A. thaliana* [18]. Therefore, it is necessary to explore the specific functions of *TCP*s in orchids.

As the largest family of monocotyledons, Orchidaceae account for 10% of total angiosperms worldwide. Therefore, they are ideal for studying biodiversity and evolution. Orchids have specialized flower organs, mainly including three sepals, two petals, one highly specialized lip, and a gynoecium [19]. *Dendrobium*, the second largest genus of Orchidaceae, contains approximately 1450 species, the majority of which are characterized by a fleshy stem with abundant polysaccharides and grow in diverse habitats [20]. *TCP*s have been widely reported in model plants and main crops; some studies have also explored the roles of *Dendrobium TCP*s in response to phytohormones by treating samples with plant hormones like methyl jasmonate (MeJA), abscisic acid (ABA), and salicylic acid(SA) [21]. Furthermore, some have conducted research on plant habits and photosynthesis pathways, investigating the diversity of biological functions of *TCP*s in *Dendrobium* species [22]. However, no studies have explored the regulating functions of *TCP*s in different flower development stages among *Dendrobium* species. With the development of sequencing technology, some orchid genomes have been reported, such as *D. chrysotoxum* [23], *D. nobile* [24], *D. huoshanense* [25], *Dendrobium officinale* [26], and *Phalaenopsis aphrodite* [27], and this provides a basis for orchid TCP gene family analysis.

In this study, a gene structure analysis, phylogenetic tree construction, collinearity analysis, and a cis-acting element analysis of the TCP gene family in three *Dendrobium* species were carried out. Specially, the expression patterns of five different floral parts in three different development stages (unpigmented bud, pigmented bud, and half-opened flower) of *D. chrysotoxum* were analyzed for the first time. Our results provide new insights into the molecular mechanisms of floral organ development and morphological diversity in *Dendrobium*. Exploring the regulation function of the TCP gene family in the flower organ of different development stages of *Dendrobium* is of great significance for the breeding and development of orchid resources. This study’s results may also help to broaden our understanding of the roles of *TCP*s in regulating flower development and floral organ determination.

## 2. Results

### 2.1. Identification and Physicochemical Properties of the TCPs

In this study, we identified the TCPs of three *Dendrobium* species based on the results of Blast and HMMER. A total of 15, 23, and 12 *TCP*s were identified in *D. chrysotoxum*, *D. nobile*, and *D. huoshanense*, respectively. In total, 50 *TCP*s were named *DchTCP1–15*, *DnoTCP1–23*, and *DhuTCP1–12* according to their distribution order on chromosomes. Furthermore, the EXPASY online tool was used to analyze the physicochemical properties of *Dendrobium*. The deduced protein length of *TCP*s varied considerably in the number of amino acids (AA), ranging from 106 aa (*DhuTCP9*) to 786 aa (*DnoTCP15*, *DhuTCP3*), and the average length was 355aa. The theoretical isoelectric point (pI) ranged from 5.62 (*DhuTCP9*) to 9.86 (*DchTCP1*). The isoelectric points of 15 TCP proteins were less than 7.0, which was acidic, and the isoelectric points of the remaining 35 *TCP*s were more than 7.0, which was alkaline. The molecular weight (MW) of 50 *TCP*s in *Dendrobium* ranged from 11.41 (*DhuTCP9*) to 88.54 kDa (*DhuTCP3*), with an average MW of 38.84 kDa. The deducted grand average of hydrophilic values (GRAVY) of the *TCP*s ranged from −0.792 to −0.118, and the whole *TCP*s showed GRAVY values less than 0, suggesting that all the TCPs were hydrophilic. The aliphatic index (AI) ranged from 58.63 (*DchTCP12*) to 84.73 (*DnoTCP6*), and the instability index (II) ranged from 38.49 (*DchTCP11*) to 74.13 (*DchTCP8*). The physicochemical properties of 50 *TCP*s in three *Dendrobium* species are listed in Appendix A.

### 2.2. Phylogenetic Analysis of TCPs

To analyze the evolutionary relationship of *TCP*s in *Dendrobium*, a phylogenetic tree consisting of *TCP*s was constructed from four species, including 24 *AthTCP*s, 15 *DchTCP*s, 23 *DnoTCP*s, and 12 *DhuTCP*s. A total of 74 *TCP*s were divided into two clades: Class-I (PCF genes) and Class-II (CYC/TB1 genes and CIN genes) (Figure 1). There were more members in Class-II than Class-I, and in Class-I, while there were more members of the CIN subfamily than there were in the CYC/TB1 subfamily, the number of Class-II members in *D. huoshanense* was similar to the other two *Dendrobium* species, but there were significant differences in Class-I (a total of 15 members; 3 *DchPCF*s, 11 *DnoPCF*s, and 1 *DhuPCF*).

### 2.3. Gene Structure and Motif Analysis of TCPs

The conserved protein motifs of the 50 TCP proteins in *Dendrobium* were predicted using the MEME online tool, and 10 motifs were set as the upper bound. The results showed that most conserved motifs existed in the C-terminal domain, and the order was Motif2, Motif1, Motif4, Motif8, Motif5, Motif6, Motif3, Motif7, Motif10, Motif9 (Figure 2A). Motif1 was the most conserved, Motif 6 existed only in CIN, and Motif7 was unique to Class-II. Almost all PCF proteins included Motif1 and Motif3; most CIN proteins included Motif2, Motif1, Motif4, and Motif5, and the CYC/TB1 proteins included Motif1, Motif2, and Motif7. This suggests that Class-II proteins contained more Motifs than Class-I proteins. To obtain the sequence logo of the TCP domains in *Dendrobium*, we conducted multiple sequence alignments. The results showed that the TCP domain encoded by Motif1 was highly conserved (Figure 2C).

To further study the characteristics of *TCP*s in *Dendrobium*, we analyzed the intron-exon structure. The results showed that all 50 *TCP*s had introns that ranged from zero to two, and exons ranged from one to three (Figure 2B). Up to 80% of *TCP*s had no introns. The longest intron appeared at *DchTCP1*, followed by *DchTCP14*. In addition, the number of exons in the Class-I genes was less than that in the Class-II genes, and *Dendrobium* exhibited fewer exon numbers than other orchids [28]. Due to differing intron lengths, the lengths of the genes among the CIN lineage were different. For example, the length of the encoded proteins of *DchTCP13* and *DchTCP14* were similar (324aa and 323aa), but the gene lengths were about 1.2 kb and 12 kb. Therefore, it is speculated that the CIN lineage might play an important regulatory role in *Dendrobium*’s growth, development, and adaptability to the external environment at the levels of gene copy number and gene structure.

### 2.4. Chromosomal Localization of TCPs

Based on the relevant genome annotation files, the gene location on the chromosomes of *D. chrysotoxum*, *D. nobile*, and *D. huoshanense* were examined. *DchTCP15* was localized to the unanchored scaffold, and the remaining fourteen *DchTCP*s were distributed on nine chromosomes. In addition, a pair of gene tandem repeats existed on Chr14 (*DchTCP9*, *DchTCP10*) (Figure 3A). A total of 23 *DnoTCP*s were distributed on 13 chromosomes of *D. nobile* (Figure 3B). Three chromosomes had three genes, four chromosomes had two genes, and the remaining six chromosomes only contained one gene. Moreover, the distribution of *DnoTCP*s was similar at the top, middle, and bottom of the chromosomes. Twelve *DhuTCP*s were distributed on nine chromosomes of *D. huoshanense* (Figure 3C). Only two *DhuTCP*s (*DhuTCP1*, *DhuTCP8*) were localized on the bottom of chromosomes, and a pair of gene tandem repeats existed on Chr16.

### 2.5. Cis-Acting Regulatory Elements Analysis of TCPs

To explore the regulatory function of *Dendrobium*, the 2000 bp promoter regions of the *DchTCP*s were retrieved to identify putative *cis*-regulatory elements. In total, 321, 499, and 226 promoter elements were identified in *D. chrysotoxum* (Figure 4A), *D. nobile* (Figure 4B), and *D. huoshanense*, respectively (Figure 4C). The functions included phytohormone responsiveness, stress responsiveness, plant growth and development responsiveness, and light response. *Cis*-acting regulatory elements related to light response were the most frequently occurring elements in *Dendrobium*, followed by MeJA-responsive and ABA-responsive elements, thereby indicating that the regulation of TCP functions by light is particularly important during plant growth and development. However, the wound-responsive element was only found in *DhuTCP12*, and the auxin-responsive element was only observed in *DhuTCP1* (Figure 4C). The top three *Dendrobium TCP*s with the highest number of *cis*-acting regulatory elements were as follows: *DnoTCP7* (34), *DnoTCP8* (33), and *DhuTCP12* (32). We also discovered meristem expression elements in three *Dendrobium* species, indicating that *TCP*s play an important role in meristem tissue.

### 2.6. Collinearity Analysis of TCPs

Six pairs of segmentally duplicated genes occurred in *D. chrysotoxum* genome (Figure 5A). Significantly, a pair of tandemly repeated genes (*DchTCP9* and *DchTCP10*) was observed on Chr14. The *D. nobile* genome showed four pairs of segmentally duplicated genes (Figure 5B). Interestingly, one pair of segmentally duplicated genes was present in the same chromosome (CM039726.1). However, only two pairs of segmentally duplicated genes were present in the *D. huoshanense* genome (Figure 5C), which also contained a pair of tandemly repeated genes (*DhuTCP6* and *DhuTCP7*) on Chr16.

### 2.7. Expression Patterns of TCPs in D. chrysotoxum

To further understand the expression patterns of the *TCP*s, we drew a heatmap of the *TCP*s in three different parts of *D. chrysotoxum* at different developmental stages (Figure 6). We found that *DchTCP12* was not detected in the transcriptome of *D. chrysotoxum*. *DchTCP1*, *DchTCP2*, and *DchTCP13* showed similar expression patterns during three stages, all belonging to the CIN subfamily. *DchTCP6, 7, 10, 11,* and *14* exhibited high expression in all three stages, but no expression was observed in the ovary in S1. *DchTCP4* represented high expression levels in the petals and lips in S1. However, *DchTCP3* showed high expression only in the ovary and sepal in S1. *DchTCP5* belonging to CYC/TB1 subfamily only showed high expression in the gynostemium in S1. Interestingly, *DchTCP8* and *DchTCP15* pertaining to the PCF subfamily were mainly expressed in S3. The expression levels of *DchTCP15* were higher than *DchTCP8*, while *DchTCP15* exhibited high expression patterns in the ovary in S1.

### 2.8. The qRT-PCR Analysis of TCPs in D. chrysotoxum

The results regarding the expression patterns showed that the expression levels of Class-II genes in the five flower parts were generally higher than that of the Class-I genes. To further understand the expression patterns of the two classes in different flower parts through the different stages of flower development, *DchTCP2* (Class-I), *DchTCP4* (Class-II), and *DchTCP13* (Class-I) were selected for qRT-PCR analysis. The results showed a relatively significant correlation between the expression trend of the selected genes and the expression level of the transcriptome data (Figure 7). *DchTCP2* and *DchTCP13* were expressed in all flower parts except the ovary in S1, and the expression levels in S1 were significantly higher than those in S2 and S3. In addition, *DchTCP4* represented no expression in the sepal in S1; the high expression in the sepal in S2 increased, and downregulation was observed in the sepal in S3. The highest expression of *DchTCP4* was observed in the petal in S1, though this gradually declined in S2 and S3.

## 3. Discussion

Orchidaceae is one of the most precious wild plant resources in the world, and the *Dendrobium* genus is particularly important due to its high ornamental value and medicinal value [29]. TCP proteins are plant-specific TFs that play a critical role in regulating plant growth and development. Although *TCP*s have been found in plants such as *Z. mays*, *O. sativa*, and *A. majus*, insufficient research has been conducted on their specific roles in orchids [30,31,32]. In this study, we identified fifty *TCP*s in three *Dendrobium* species. Our study’s results showed that the number of *TCP*s in *D. chrysotoxum* (15) and *D. huoshanense* (12) were similar to *C. goeringii* (14) [33] but less than *Phalaenopsis equestris* (23) [34] and *D. catenatum* (25) [35]. *D. chrytosoxum* and *D. huoshanense* may have suffered genetic loss during evolution. This indicates that genome size, chromosome numbers, and gene duplication are potentially responsible for the differences.

The TCP gene family was divided into two classes including three subfamilies (PCF, CIN, and CYC/TB1) based on a phylogenetic tree that pertained to *A. thaliana*, *D. chrysotoxum*, *D. nobile*, and *D. huoshanense*. The phylogenetic analysis indicated that only *AtTCP16* was not found to have a similar homolog. The number of *TCP*s in Class-II was similar among the three *Dendrobium* species, but there was a significant difference in the number of *TCP*s in Class-I (3 *DchPCF*s, 11 *DnoPCF*s, and 1 *DhuPCF*). The number of amino acids in the *TCP*s of three *Dendrobium* species demonstrated a significant difference, ranging from 106 aa (*DhuTCP9*) to 786 aa (*DnoTCP15*, *DhuTCP3*). These results showed the potential complexity of the origin and evolution of the *Dendrobium* TCP gene family, suggesting a strong relationship with biological functional diversity [36].

Our gene structure analysis indicated that the three *Dendrobium* species had highly similar exon–intron structures. Almost all reported orchids have long introns, but only three long introns were observed in this study. Interestingly, they were all displayed in *D. chrysotoxum*. *C. goeringii* only has four long introns [33]. The genes with introns only account for 20% of all 50 *TCP*s in *Dendrobium*, possibly due to intron loss during gene structure evolution [37]. Despite not providing any benefit in recombination nor species evolution, intron-free genes frequently react quickly to stress [38]. Furthermore, we analyzed the conserved motifs in three *Dendrobium* species. The Class-II genes containing Motif1, 2, and 4 were more conservative than the Class-I genes. Motif 6 only showed up in CIN subfamilies, suggesting that CIN genes may have particular biological functions. Through the Motif logo in three *Dendrobium* species and the above analysis, it can be concluded that the TCP domain of *Dendrobium* is relatively conserved in the plant evolutionary process.

Whole genome duplication (WGD) [39,40,41] is considered an important factor in genome evolution and gene family formation and expansion [42,43]. *D. chrysotoxum*, *D. nobile,* and *D. huoshanense* have experienced at least two whole-genome duplication events [21,22,23]. Therefore, the *TCP* number and distribution in the three *Dendrobium* species have some differences. There was a pair of tandemly repeated genes on Chr14 in *D. chrysotoxum* and Chr16 in *D. huoshanense*, respectively. The collinearity analysis results showed six pairs of segmentally duplicated genes in *D. chrysotoxum*, four pairs of segmentally duplicated genes in *D. nobile*, and only two pairs of segmentally duplicated genes in *D. huoshanense*. Overall, these findings indicate that whole-genome duplication plays a role in the expansion of the TCP gene family of *Dendrobium*.

The prediction of promoter elements at the transcriptional level provides a more thorough understanding of gene regulation [44]. The *Cis*-acting elements for a series of functional types were identified in the *TCP*s of three *Dendrobium* species, including light-responsive, phytohormone-responsive, stress-responsive, and plant growth and development elements. In this study, we found that the light-responsive elements were the most frequently occurring elements in the 2000 bp upstream sequences from *Dendrobium*, accounting for 41.3% of all *cis*-acting elements. Light affects multiple developmental processes, including seed germination, shade avoidance, and photoperiod responses [45]. So, our analysis indicated that *TCP*s may play a role in these developmental processes. The *cis*-acting elements of phytohormone accounted for 16.1% of all elements, mainly including MeJA. MeJA is a vital cellular regulator against biotic and abiotic stresses in different development processes [46]. Therefore, it is speculated that *TCP*s may impact the stress resistance of *Dendrobium*. The proportion of biotic stresses and growth and development elements were 16.0% and 26.6%, respectively. More *cis*-acting elements were related to meristem expression in the last type. The identification of *TCP1* in the regulation brassinosteroid (BR) biosynthesis suggested that *TCP1* may cause the uneven distribution of BRs in floral meristems [47]. In the inflorescence shoot apex and seed, DELLA proteins release *TCP*s to simulate cell division in the root apical meristem, thereby separately stimulating shoot elongation and seed germination [48,49]. Based on the above discussion, we can infer that *Dendrobium TCP*s is actively involved in meristem development.

It has been reported that TCP transcription factors are involved in plant development processes such as gametophyte development, lateral branching, seed germination, flower development, and leaf development [1,2,3,4,5,6,7,8,9,10]. Through expression analysis, we found that, among all of the detected *DchTCP*s, the expression profiles of the Class-II genes were more active than that of the Class-I genes. Moreover, the DNA sequences recognized by Class-I or Class-II *TCP*s are not mutually exclusive, suggesting that they may share the same target genes and that they function together in regulating gene expression [50]. It would be interesting to determine if Class-I and Class-II *TCP*s interact with the same target gene in *Dendrobium*. Moreover, the expression levels of *TCP*s from the CYC/TB1 subfamily were lower than those from the CIN subfamily, and *DchTCP12* from the CYC/TB1 subfamily was not detected in the transcriptome. *DchTCP4* and *DchTCP15* from the PCF subfamily showed active expression levels in the gynostemium and ovary in the unpigmented bud stage, further confirming the findings of a previous study that reported that PCFs in the TCP gene family regulate seed germination and gametophyte development [51]. Interestingly, *DchTCP2* and *DchTCP13* participated in all parts in S1 except the ovary, and their expression levels in S1 were much higher than those of the other two stages. Based on these results, we speculate that *DchCIN*s may be involved in the color transformation process of flower buds. In addition, these two genes were expressed in both sepal and petal in the S1 and S2, which is consistent with the function of *CIN*s participating in the development of lateral organs [52,53]. It has been reported that the TCP gene BRC1 interacts with flowering genes and modulates floral activity in the axillary buds [54]. Furthermore, the decrease in five Class-II *TCP*s may be related to the delayed flowering phenotype of *Arabidopsis* mutants [55]. We speculate that *DchTCP2* and *DchTCP13* may exert an important effect on the regulation of florescence. Previous studies have found that the downregulation of *AtTCP14*, *15* from the PCF subfamily may affect the development of outer whorls and gynoecia [56]. Combining the downregulation of *DchTCP4* during three stages, we propose that *DchTCP4* belonging to PCF subfamily may have a major effect on the phenotype of floral organs.

## 4. Materials and Methods

### 4.1. Plant Materials

The plant materials were collected from the National Orchid Germplasm Resources of Fujian Agriculture and Forestry University, Fuzhou, China. We used five different parts of *D. chrysotoxum* (petal, sepal, lip, ovary, and gynostemium) at three developmental stages, namely the unpigmented bud stage (S1), the pigmented bud stage (S2), and the half-opened flower stage (S3). All samples were frozen at −80 °C.

### 4.2. Identification and Physicochemical Properties of the TCP Gene Family

In order to identify potential *TCP*s in *Dendrobium*, 24 *A. thaliana* TCP proteins were used as a probe in a local BLASTp search (built-in TBtools v 1.120) [26]. The Hidden Markov Model (HMM) profile of the TCP conserved domain (PF03634) from the online database (http://pfam-legacy.xfam.org/, accessed on 13 June 2023) was used to further identify *TCP*s in *Dendrobium* using the Simple HMM Search in TBtools. The results of BLASTp and HMMER were combined to manually remove incomplete and redundant protein sequences, and uncertain genes were uploaded to the NCBI (http://blast.ncbi.nlm.njh.gov/, accessed on 13 June 2023) website prior to a BLASTp search. The online software ExPASy (http://www.expasy.org/, accessed on 13 June 2023) was used to analyze the amino acid (AA) content, isoelectric points (pIs), molecular weights (MWs), hydrophilic large averages (GRAVY), instability index (II), and fat index (AI) of the proteins [27].

### 4.3. Phylogenetic Analysis of TCPs

The 24 TCP protein sequences of *A. thalian* (*AthTCP*), 23 TCP protein sequences of *D. nobile* (*DnoTCP*), 12 TCP protein sequences of *D. huoshanense* (*DhuTCP*), and 15 TCP protein sequences of *D. chrysotoxum* (*DchTCP*) were introduced into the MEGA7.0 software. The alignment sequences selected that were using the ClustalW program for multisequence alignment to assess the similarities between proteins, Gap Opening and Gap Extend, were 15 and 6.66, respectively; the DNA Weight Matrix selection was the IUB; other values were kept as the default values. Then, the neighbor-joining (NJ) method was carried out using 500 replications of the bootstrap method to construct a phylogenetic tree consisting of a total of 50 sequences. The percentage of replicate trees in which the associated taxa clustered together in the bootstrap test (500 replicates) is shown next to the branches. In addition, we performed the bootstrap method through using 500 replicates (partial deletion with 50%). The online software Evloview3.0 (https://evolgenius.info//evolview-v2/#mytrees/tree/ATDc%20TCP, accessed on 15 June 2023) was used to improve and beautify the phylogenetic tree.

### 4.4. Conserved Motifs and Gene Structure Analysis of TCPs

The conserved domains of the *TCP*s were analyzed using the NCBI conserved domain database (CDD) (https://www.ncbi.nlm.nih.gov/Structure/bwrpsb/bwrpsb.cgi, accessed on 13 June 2023). For conserved motif prediction, all TCP protein sequences of *Dendrobium* were submitted to the Multiple Em for Motif Elicitation (MEME) online tool (http://meme-suite.org/meme/tools/meme, accessed on 13 June 2023) for identification. We set the maximum number of motifs at ten, and the other parameters were kept at the defaults. Finally, the phylogenetic tree, conserved motifs, and gene structures of the *Dendrobium TCP*s were visualized using TBtools.

### 4.5. Collinearity Analysis and Chromosomal Localization of TCPs

Using the Gene Location Visualize tool from the GTF/GFF programs in TBtools with the annotation files for *Dendrobium*, we obtained the chromosomal locations of the *Dendrobium TCP*s. For collinearity analysis, the genome data of three *Dendrobium* species were compared to themselves via the use of the One Step MCScanx program in TBtools. Then, we used the One Step MCScanx program and the Advance Circos to visualize the duplication patterns in three *Dendrobium* species.

### 4.6. Promoter Element Analysis of TCP Genes

We used TBtools to obtain a 2000 bp gene sequence upstream of the promoter codon from the genome of three *Dendrobium* species to identify putative *cis* elements in the promotor [26]. PlantCARE online software (https://bioinformatics.psb.ugent.be/webtools/plantcare/html/, accessed on 20 June 2023) was used to analyze *cis*-regulatory elements in the promoter region of the *TCP*s in *Dendrobium*. Finally, the data were processed and appropriately sorted using Excel 2016 software before being visualized using TBtools v2.003.

### 4.7. Expression Analysis and RT-qPCR

Transcriptome sequencing and library construction were completed by Bgi Genomics Co., Ltd. (Shenzhen, China). Bowtie2 2.2.9 was used to compare the clean reads to the genomic sequence and RSEMv1.2.8 software was used to calculate the gene expression level of each sample. Gene expression levels were measured as fragments per kilobase of transcript per million fragments (FPKM) values. The data were visualized using TBtools v2.003, and a heat map was generated for *D. chrysotoxum*.

Total RNA was extracted from the plant materials using a FastPure Plant Total RNA Isolation Kit (for polysaccharide- and polyphenol-rich tissues) (Vazyme Biotech Co., Ltd., Nanjing, China). A reverse Transcript Kit PrimerScript^®®^ RT reagent Kit with gDNA Eraser (TaKaRa, Dalian, China) was used for reverse transcription to remove the contaminated genomic DNA and generate cDNA. Primer Premier 5 software was used to design primers, and *Maker75111* was selected as the reference gene based on the transcriptome data (Appendix A). Hieff UNICON^®^ Universal Blue qPCR SYBR Green Master Mix was used for a qRT-PCR analysis on an ABI 7500 Real-Time System. The experimental setup utilized 96-well plates with a 20 µL reaction system in each well, and three biological replicates of the study were performed. The RT-qPCR conditions were 2 min at 95 °C in the holding stage and 40 cycles of 10 s at 95 °C and 30 s at 60 °C in the cycling stage. The relative expression levels of the target genes were calculated using the 2^−ΔΔCT^ method for the experimental data and normalized using GraphPad Prism 7.0 [57].

## 5. Conclusions

For this study, we identified 15 *DchTCP*s, 23 *DnoTCP*s, and 12 *DhuTCP*s in three *Dendrobium* species and classified them into two clades—Class-I (PCF subfamily) and Class-II (CIN and CYC/TB1 subfamilies)—based on their phylogenetic relationships. Analyses of phylogenetics, gene structure, motif composition, chromosomal localization, and *cis*-acting elements in the three *Dendrobium* species were carried out. In addition, the results of our expression and qRT-PCR analyses indicated that *DchTCP2* and *DchTCP13* have a certain impact on the lateral organs, and changes in the expression level of *DchTCP4* may affect the phenotypes of floral organs. In summary, this study provides a reference for researchers to better understand how *TCP*s are involved in regulating flower development and floral organ determination.

## Figures and Tables

**Figure 1 ijms-24-14320-f001:**
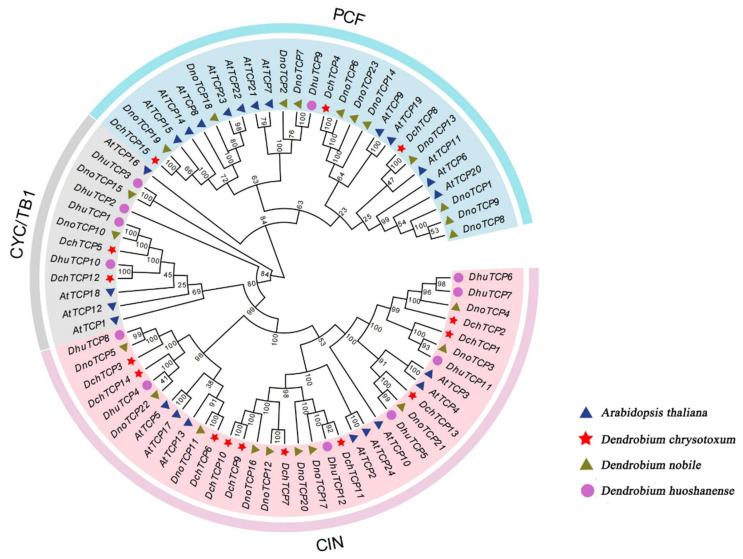
**Phylogenetic tree of 74 TCP proteins from *A. thaliana*, *D. chrysotoxum*, *D. nobile*, *D. huoshanense*.** The phylogenetic tree was constructed using the neighbor-joining (NJ) method in MEGA7.0 software and was divided into two classes (three subfamilies) according to the classification of *AtTCP*s.

**Figure 2 ijms-24-14320-f002:**
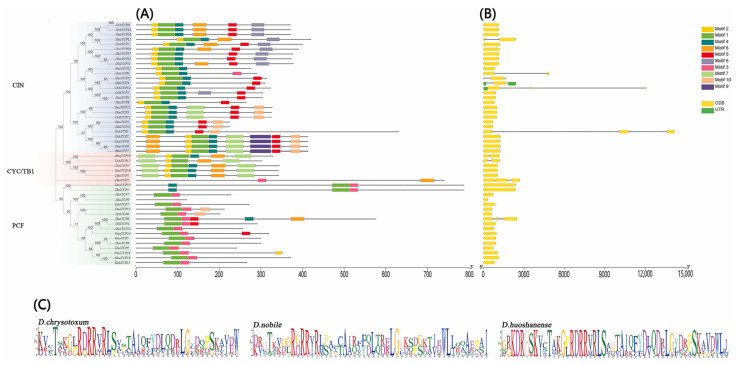
**Conserved motifs, gene structure, and conserved domains of *TCP*s.** (**A**) Phylogenetic tree and conserved motifs of the *TCP*s of *D. chrysotoxum*, *D. nobile*, and *D. huoshanense*. (**B**) Gene structure of the *TCP*s of *D. chrysotoxum*, *D. nobile*, and *D. huoshanense*. (**C**) Sequence logo of Motif1 of *D. chrysotoxum*, *D. nobile*, and *D. huoshanense*, which encodes the TCP domain.

**Figure 3 ijms-24-14320-f003:**
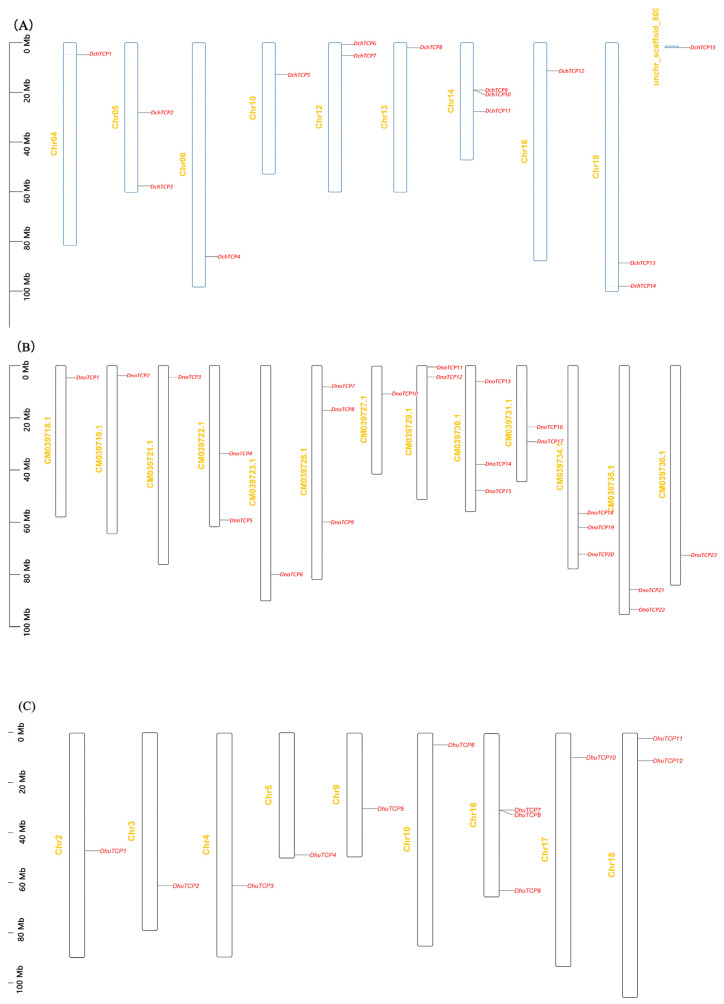
**Chromosome distribution in three *Dendrobium* species.** (**A**) Chromosome distribution in *D. chrysotoxum*. (**B**) Chromosome distribution in *D. nobile*. (**C**) Chromosome distribution in *D. huoshanense*. The yellow letters represent chromosome names, while the red letters represent gene names.

**Figure 4 ijms-24-14320-f004:**
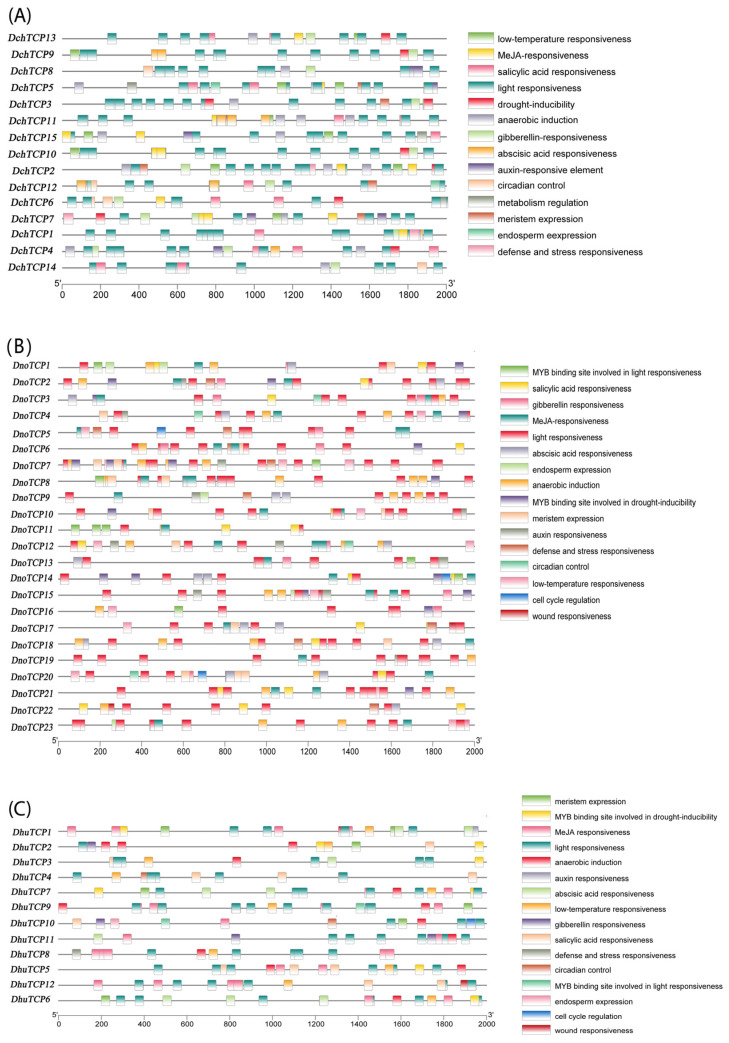
**Regulatory elements in the promotor region of three *Dendrobium* species.** (**A**) The *cis*-acting elements of *D. chrysotoxum*. (**B**) The *cis*-acting elements of *D. nobile*. (**C**) The *cis*-acting elements of *D. huoshanense*. Raw data are listed in Appendix A.

**Figure 5 ijms-24-14320-f005:**
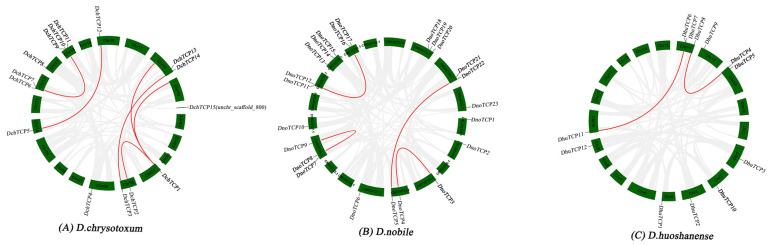
**Synteny analysis of *TCP*s in three *Dendrobium* species.** (**A**) Synteny analysis of *DchTCP*s. (**B**) Synteny analysis of *DnoTCP*s. (**C**) Synteny analysis of *DhuTCP*s. Red lines represent segmentally duplicated gene pairs.

**Figure 6 ijms-24-14320-f006:**
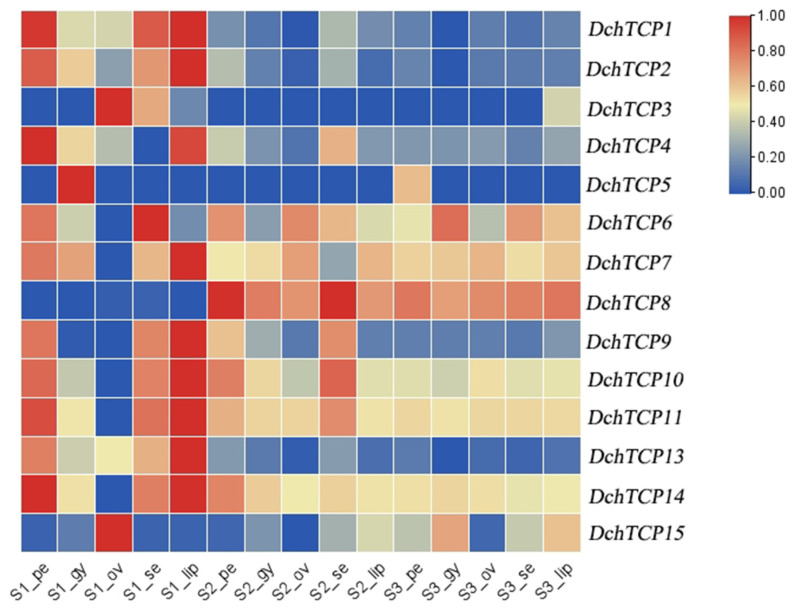
**The expression levels of 14 *DchTCP*s in different parts and developmental periods in *D. chrysotoxum*.** S1: unpigmented bud stage; S2: pigmented bud stage; S3: half-opened flower stage; pe: petal; se: sepal; lip: lip; gy: gynostemium; ov: ovary. The FPKM of *TCP*s in *D. chrysotoxum* are listed in Appendix A.

**Figure 7 ijms-24-14320-f007:**
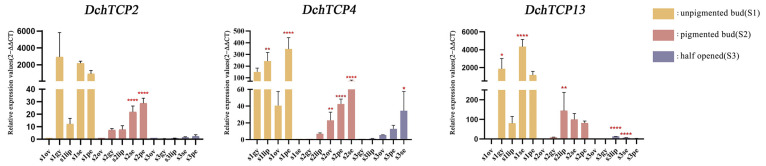
Real-time fluorescence quantitative expression analysis on *DchTCP*s at the different stages of flower development. Raw data are listed in Appendix A. Y-axis represents relative expression values (2−ΔΔCT). The red asterisk indicates the P value in the significance test (* *p* < 0.05, ** *p* < 0.01, **** *p* < 0.0001). Primers and RT-qPCR analyses of TCP genes are shown in Appendix A.

## Data Availability

The genome sequence annotation files of *D. chrysotoxum* and *D. nobile* were downloaded from the National Center for Biotechnology Information (NCBI, https://www.ncbi.nlm.nih.gov/, accessed on 12 June 2023) (accession number: PRJNA664445, PRJNA725550), the *D. huoshanense* files were downloaded from the China Nucleotide Sequence Archive (CNSA, https://ftp.cngb.org/, accessed on 12 June 2023), and the TCP protein sequence files of *A.thaliana* were downloaded from TAIR (http://www.arabidopsis.org, accessed on 12 June 2023) and EnsemblPlants (http://plants.ensembl.org/index.html, accessed on 12 June 2023), respectively.

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
