# Peer review of "Genome-Wide Identification of TCP Gene Family in Dendrobium and Their Expression Patterns in Dendrobium chrysotoxum"

_ijms, 2023, doi:10.3390/ijms241814320_

Round 1

Reviewer 1 Report

The study was focused on genome-wide identification of TCP gene family in Dendrobium and their expression patterns in Dendrobium chrysotoxum. Authors have identified a total of 50 TCPs across three Dendrobium species, including 15 DchTCPs, 23 DnoTCPs, and 12 DhuTCPs. They were divided into two clades, Class-(PCF subfamily) and Class-(CIN and CYC/TB1 subfamilies), based on the phylogenetic results. The sequence analysis showed that almost all Dendrobium TCPs contain conserved TCP domain, and gene structure analysis suggested the existence of fewer exons in Dendrobium compared with other orchids, while most TCPs had no introns. The results of cis-elements of TCPs were mostly related to light response and phytohormone regulation.

The paper is quite interesting, however, I recommend some significant major improvements:

-        Abstract and Introduction are overloaded in content.

-        I recommend including the electropherograms presenting the RNA bands in agarose gels in the manuscript or in the Supplementary file – it would provide information regarding quality of total RNA samples. The RNA concentration should be quantified at both 260 and 280 nm, and purity of RNA should be calculated based on the A260/A280 coefficient.

-        Authors used SYBR Green fluorescent dye during RT-PCR gene expression studies, hence, it is obligatory to perform Melting Curve Analysis, and results of this examination should be added in the manuscript or Supplementary file (e.g., JPG or TIFF file).

-        Discussion part should be significantly deepened, and extended. At present, the discussion is plain/superficial.

-        There is lack of citation of double delta Ct method used for gene expression analyses (Livak and Schmittgen 2001).

-        There is no information regarding statistical analyses, it should be calculated and described in the manuscript, as well as statistical significance should be marked above the figures.

-        Extensive editing of English language is required.

Extensive editing of English language is required.

Reviewer 2 Report

Manuscript "Genome-wide identification of TCP gene family in Dendrobium and their expression patterns in Dendrobium chrysotoxum" is very interesting.

General comments:
Authors identified TCPs in three Dendrobium species, and then performed comprehensive bioinformatics analyses of phylogenetic tree, gene structure, cis- element, chromosomal location, and segmental duplication events. Authors used transcriptome sequencing to study the expression patterns of TCPs genes in D. chrysotoxum.

Detailed comments:
The manuscript was prepared in a sloppy manner.
When quoting: insert a space before "[". Be corrected throughout the manuscript.
Authors used five different parts of D. chrysotoxum (petal, sepal, lip, ovary, and gynostemium) at three developmental stages, including unpigmented bud (S1), pigmented bud (S2), and half opened flower (S3). Unfortunately, a two-way analysis of variance is lacking.
The Authors used the Hidden Markov Model; unfortunately, they did not present the probabilistic parameters of the model. It should be supplemented.
Figure 1: information on the method used to assess similarity between proteins is missing.
Figure 2A: information on the method used to assess similarity between proteins is missing.
Figure 7: the statistical analysis of the data presented here is missing. It is necessary to complete.

Paper needs major revision.

Round 2

Reviewer 1 Report

The revision was properly concucted. The manuscript may be considered for publication. 

 Minor editing of English language is required.

Reviewer 2 Report

Comments 3: Authors used five different parts of D. chrysotoxum (petal, sepal, lip, ovary, and gynostemium) at three developmental stages, including unpigmented bud (S1), pigmented bud (S2), and half opened flower (S3). Unfortunately, a two-way analysis of variance is lacking.

Response 3: Thank you for your comments. We have marked it above the figures. Please see Figure 6.

Re: The results of the two-way analysis of variance are still missing. To be completed.

Comments 5: Figure 1: information on the method used to assess similarity between proteins is missing.

Response 5: Thank you for your comments. We have revised the figure. Please see Figure 1.

Re: Despite responding that they had revised the manuscript, the authors still did not provide information about the method used to calculate the similarity between the proteins.

Comments 6: Figure 2A: information on the method used to assess similarity between proteins is missing.

Response 6: Thank you for pointing this out. We have made adjustments to the figure. Please see Figure 2.
Re: Despite responding that they had revised the manuscript, the authors still did not provide information about the method used to calculate the similarity between the proteins.

Paper needs revision.

Round 3

Reviewer 2 Report

Authors have incorporated all the suggestions, accordingly. I recommend this article to publish in current version.

Author Response

Thank you for your patience and energy on this manuscript, which enables us to complete the revision of our manuscript faster.